# OpenReview forum: "Lottery Ticket Adaptation: Mitigating Destructive Interference in LLMs"
_ICLR.cc/2025/Conference — ICLR 2025 Conference Withdrawn Submission_

### Official Review · Reviewer_WcFw · 2024-10-22

**Soundness:** 3
**Presentation:** 3
**Contribution:** 3
**Rating:** 6
**Confidence:** 4

**Summary:**

The authors propose LoTA, an interesting framework that solving the problem of three different type of methods about multitask adaptation. LoTA is a very simple but effective framework to implement the multitask learning compare to previous methods.

**Strengths:**

- The method LoTA is very simple but effective, successfully solving the problems of previous methods.

- Empirical results, especially on Model merging and sequential training setting.

- The paper is easier to follow.

**Weaknesses:**

- The paper only uses magnitude-based pruning methods.

- It will be more fair to compare FFT and LoTA if FFT have the same number of epochs as the sum number of LoTA in the first and third phase.

- The additional phase may introduce additional computation cost (although it is not very much)

- Compare to LoRA and other PEFT methods, LoTA requires more GPU memory requirement when training.

**Questions:**

- Why choosing an FFT then sparse framework rather than select-updating-parameter, which means first select some parameter can be updated than train the model like the process in Sparse Adaptation?

- Is it possible to sharing the results for many task settings? For example, in all the task in BBH or MMLU benchmark. The experiments in only two tasks is not very persuasive. If possible, I guess the most important results is the gap between your methods and directly mix all the data to FFT.

I may raise the score if all questions are solved.

---

> ### Author Response · Authors · 2024-11-18
>
> Thank you for the detailed review. We are still running some experiments, but have some initial clarifications to your questions (just pointing you to the line number in the paper where we did the thing you asked for).
>
> > It will be more fair to compare FFT and LoTA if FFT have the same number of epochs as the sum number of LoTA in the first and third phase.
>
> We agree! Line 299: "We run all the methods for the same number of steps, i.e., the calibration time for LoTA is factored in."
>
> > If possible, I guess the most important results is the gap between your methods and directly mix all the data to FFT.
>
> We agree! Line 373: "Sequential Training Under LoTTO Outperforms Mixing Datasets Together." is the paragraph describing this experiment, and the results are in Table 5.
>
> > Only magnitude-based pruning
> > Why not compare to Sparse Adaptation
> > Results for many task settings
>
> We are working on these and will aim to have it by the end of the rebuttal window!

---

### Official Review · Reviewer_Z8vz · 2024-10-30

**Soundness:** 2
**Presentation:** 3
**Contribution:** 2
**Rating:** 3
**Confidence:** 4

**Summary:**

This work introduced Lottery Ticket Adaptation (LoTA), a sparse adaptation method that optimizes only a subnetwork of the model to avoid destructive interference between tasks. Unlike existing methods that often cause catastrophic forgetting and performance drops, LoTA maintains high performance across multiple tasks by focusing on sparse task vectors. As a result, LoTA outperforms full fine-tuning and low-rank adaptation (LoRA) in various tasks and enables effective model merging without performance degradation.

**Strengths:**

This paper introduces a new method called Lottery Ticket Adaptation (LoTA). LoTA works by identifying and optimizing sparse task vectors, which means it only tweaks a small, crucial part of the model. This selective tuning helps the model remember its core skills while learning new tasks, thus preventing the common problem of catastrophic forgetting.

The authors thoroughly test their method on various challenging tasks, such as following instructions, reasoning, solving math problems, and summarizing text. These tests are conducted on large language models, showing that LoTA is not just a theoretical idea but works well in real-world scenarios. The experiments prove that LoTA performs better than traditional methods like full fine-tuning and low-rank adaptation (LoRA).

One of the standout features of LoTA is its ability to maintain high performance even after the model has been trained on multiple tasks. Unlike other methods that often cause the model to forget previously learned skills, LoTA ensures that the model remains robust and effective, making it a reliable choice for multi-task learning.

**Weaknesses:**

See Questions Section

**Questions:**

Why does the paper not include a comparison of the operational efficiency of the LoTA method, specifically in terms of GPU memory cost and training speed? Understanding these aspects is crucial for assessing the practicality and scalability of LoTA in real-world applications.

Why were certain baselines, such as LoRAHub(LoraHub: Efficient Cross-Task Generalization via Dynamic LoRA Composition), LoRAMoE(LoRAMoE: Alleviate World Knowledge Forgetting in Large Language Models via MoE-Style Plugin), and MoLE (Mixture of LoRA Experts), not included in the experimental comparisons? These methods are also designed to address multi-task adaptation in the LoRA setting, and their inclusion would provide a more comprehensive evaluation of LoTA's performance relative to other state-of-the-art approaches.

The paper needs to provide more evidence to demonstrate the importance of avoiding catastrophic forgetting in large language models (LLMs)? It is often assumed that LLMs are not further trained after their initial post-training phase, which would imply that catastrophic forgetting is not a significant issue. Providing concrete examples or scenarios where continual adaptation or fine-tuning is necessary would help justify the relevance of methods like LoTA in this context.

---

### Official Review · Reviewer_Lqq1 · 2024-10-31

**Soundness:** 1
**Presentation:** 2
**Contribution:** 3
**Rating:** 3
**Confidence:** 4

**Summary:**

The work presents a method, LoTA, leveraging sparse vector masks to finetune a model by modifying only a subset of its weights. It allows for sequential training with multiple LoTAs on different tasks leading to lowered catastrophic forgetting - called LoTTO method. The evaluation methodology covers instruction following, commonsense reasoning, math, and summarization tasks, with comparisons to full fine-tuning and LoRA approaches.

**Strengths:**

The work leverages sparsity of model weights updates to provide a method for multi-task adaptation. It has similarities to post-hoc sparsification but allows to train these sparse weight deltas after mask calibration which differentiates it from other methods. Even if it may not have high practical utility in the current form due to high computational requirements compared to PEFT methods, the idea is interesting and may lead to significant improvements in multi-task adaptation approaches.

**Weaknesses:**

The idea is generally good, but there are a number of issues which prevent from recommending the acceptance of the paper, and that need to be resolved.

- Regarding catastrophic forgetting with LoTTO, there's lack of comprehensive experiments in regard to different task combinations to prove the robustness of the method - it was limited to forgetting of the GSM8k (Table 4) and Instruction Tuning (Table 5 & 6).
- The experiments for sequential training are limited to only two tasks/LoTAs and don't provide data on how the performance drops for every additional LoTA adapter due to more limited parameter space (as the sparse vector masks in LoTTO should not overlap).
- The computational requirements (memory & runtime) are not clearly discussed - while LoTA may be better than LoRA in terms of catastrophic forgetting and/or downstream task performance, it does not bring the same benefits in computational efficiency as PEFT methods - it's mentioned in limitations paragraph with one sentence but may need more emphasis. Given additional operations on sparse vector masks, LoTA may require even more memory & runtime compared to FFT - it's just a speculation and would need a proper comparison from the authors.
- Results in Table 4 seem surprising given that performance of FFT and LoRA completely collapses on the GSM8K task, and is in contrary to results from (Biderman et al., 2024) as mentioned in the paper. I'm suspecting it's due to different structures of the tasks - GSM8K needs open generation with EOS token at the end, for which the final answer is extracted eg. with regex, while QA (commonsense reasoning tasks) indicate correct answer based on tokens' logits, and in the simplest case it's based on a single token of a letter corresponding to a given answer. It would be good to include results from both models, swapped tasks (QA first, then GSM8k), and possibly two tasks of the same structure (either both open generation or both QA). Another issue I see is that LoRA has been trained with ranks 512 and 64, while in practice lower values are used - e.g. Biderman et al. used rank 16 for decreased forgetting.
- Arbitrary choices without explanation:
    - Introduces two models in Section 4, but shows results only for a single model except one table - for some tables it's not stated which model is used.
    - Line 249 - "As a representative task, we use ARC-easy", without any further explanation.
    - Use of RMSProp optimizer instead of Adam which is used in majority of works.

**Questions:**

Questions:
- How much % of parameters is trained on average with LoTA? How much does it vary across tasks and model scales?
- What is the performance drop for every new task trained with LoTTO? I would expect that constraining parameter space with each additional task reduced the potential performance on new LoTAs.
- What is the reason of the choice of RMSProp optimizer? Do the results differ for Adam optimizer, which is de facto a standard for NN optimization?
- Are the experiments repeated for different random seeds, and results averaged?
- Line 249, "As a representative task, we use ARC-easy." Why?
- Regarding experiments in section 5.2, is LoRA completely forgetting Task A also when using lower rank, e.g. 8? Work by Biderman et al. (mentioned at line 353) shows that lower rank leads to less forgetting. Moreover, are the results similar after swapping Task A with Task B? What are the results for Llama3-8B?

Suggestions (mostly regarding presentation):
- Line 263, "finetuning method OpenAI are using in their fine-tuning API (presumably FFT)." It's not a known fact, and in my opinion also unlikely one - such speculations should not be part of a paper.
- Table 3: Why only LoTA is bolded out, if for some tasks other methods achieve the same scores?
- Line 205, "we do not expect that LoTTO will result in any worse forgetting than LoTA" Is it a typo? Shouldn't it say LoRA?
- I would suggest to find a better label for tasks - Boolq, PIQA, SocialIQA, Hellaswag, Winograde, ARC-easy, ARC-challenge, OpenBookQA - as they are not specifically testing **reasoning** capabilities of the model and it's misleading. Natural Language Understanding (NLU) or Question Answering (QA) would be a better label in my opinion.
- Table 5 is not referred to in the text. It's also not clear what metric is used for both tasks - should be mentioned in the caption or the column header.

---

### Official Review · Reviewer_cEA4 · 2024-11-04

**Soundness:** 3
**Presentation:** 4
**Contribution:** 2
**Rating:** 5
**Confidence:** 3

**Summary:**

The paper proposes a fine-tuning method for adapting LLMs to enable multi-task learning and prevent catastrophic forgetting. The method uses a mask calibration stage that extracts important neurons based on the magnitude of the task vector and then performs a sparse update by applying the mask. The paper conducts thorough experiments on single-task adaptation, serial training, and model merging to compare the proposed method with existing methods and demonstrates the superior performance and robustness of the method across settings.

**Strengths:**

1. The paper is well-written and the proposed method is clearly presented.

2. The proposed method is evaluated on a comprehensive set of settings, including sequential training and model merging which common PEFT papers do not usually evaluate. This adds another layer of strength and depth to the paper.

3. The proposed method shows empirically strong results over full fine-tuning and LoRA across all the aforementioned settings.

**Weaknesses:**

1. My main concern is the technical novelty of the proposed method, particularly LoTA. [1] (and [2], the follow-up work by the same authors that scaled up to the latest LLMs) proposed a very similar method called "Lottery Ticket Sparse Fine-Tuning." By comparing Section 3.1 of [1] and Algorithm 1 of this paper, I find the only difference to be that [1] applies an L1 regularization for the sparse adaptation training part whereas this paper does not. To the best of my knowledge, the second algorithm proposed by the author, LoTTO, is novel as it forces the adaptations to be disjoint for different tasks; however, it would be much appreciated if the authors could clarify the main technical contributions that differentiate LoTA and prior works.

2. Regardless of 1, I think this paper lacks proper baselines of sparse/localized fine-tuning. The proposed method is only compared with full fine-tuning and dense PEFT method (LoRA). The paper can be strengthened by a lot if the proposed method can be compared with at least one sparse/localized fine-tuning methods, such as [1] or [3].

Despite the aforementioned weakness, this paper still presents a thorough suite of experiments and evaluations. I'd love to discuss with the authors and am willing to change my rating if the authors can address these main concerns.

[1] Alan Ansell, Edoardo Maria Ponti, Anna Korhonen, and Ivan Vulić. Composable Sparse Fine-Tuning for Cross-Lingual Transfer. 2021
[2] Alan Ansell, Ivan Vulić, Hannah Sterz, Anna Korhonen, and Edoardo M. Ponti. Scaling Sparse Fine-Tuning to Large Language Models. 2024
[3] Fangcong Yin, Xi Ye, and Greg Durrett. LoFiT: Localized Fine-tuning on LLM Representations. 2024

**Questions:**

1. Lines 250 -253: The authors have mentioned the issue of using potentially contaminated benchmarks such as MT-Bench and avoid using such. However, GSM8K is also one of the benchmarks that are known for potential data contamination (e.g. [1]).

2. How many training steps does it usually take for the mask calibration part of LoTA? Is it just one training step, as line 292 says"We find that a 90%-sparse mask can achieve good performance on GSM8K with just a single step of calibration"?

3. Table 4: Can you provide a bit more justification on why choosing the alternating LoRA/FFT as a baseline?

[1] Hugh Zhang, Jeff Da, Dean Lee, Vaughn Robinson, Catherine Wu, Will Song, Tiffany Zhao, Pranav Raja, Dylan Slack, Qin Lyu, Sean Hendryx, Russell Kaplan, Michele Lunati, and Summer Yue. A Careful Examination of Large Language Model Performance on Grade School Arithmetic. 2024

---

### Note · Authors · 2024-11-24

I have read and agree with the venue's withdrawal policy on behalf of myself and my co-authors.